# Mechanistic Investigation of the Formation of Nickel Nanocrystallites Embedded in Amorphous Silicon Nitride Nanocomposites

**DOI:** 10.3390/nano12101644

**Published:** 2022-05-11

**Authors:** Norifumi Asakuma, Shotaro Tada, Erika Kawaguchi, Motoharu Terashima, Sawao Honda, Rafael Kenji Nishihora, Pierre Carles, Samuel Bernard, Yuji Iwamoto

**Affiliations:** 1Department of Life Science and Applied Chemistry, Graduated School of Engineering, Nagoya Institute of Technology, Gokiso-cho, Showa-ku, Nagoya 466-8555, Japan; n.asakuma.633@stn.nitech.ac.jp (N.A.); tada.shotaro@nitech.ac.jp (S.T.); e.kawaguchi.310@stn.nitech.ac.jp (E.K.); m.terashima.311@stn.nitech.ac.jp (M.T.); honda@nitech.ac.jp (S.H.); 2CNRS, IRCER, UMR 7315, University of Limoges, F-87000 Limoges, France; rafael.nishihora@hotmail.com (R.K.N.); pierre.carles@unilim.fr (P.C.); samuel.bernard@unilim.fr (S.B.)

**Keywords:** Ni nanocrystallites, amorphous silicon nitride, nanocomposite, low temperature formation, polymer-derived ceramics

## Abstract

Herein, we report the mechanistic investigation of the formation of nickel (Ni) nanocrystallites during the formation of amorphous silicon nitride at a temperature as low as 400 °C, using perhydropolysilazane (PHPS) as a preformed precursor and further coordinated by nickel chloride (NiCl_2_); thus, forming the non-noble transition metal (TM) as a potential catalyst and the support in an one-step process. It was demonstrated that NiCl_2_ catalyzed dehydrocoupling reactions between Si-H and N-H bonds in PHPS to afford ternary silylamino groups, which resulted in the formation of a nanocomposite precursor via complex formation: Ni(II) cation of NiCl_2_ coordinated the ternary silylamino ligands formed in situ. By monitoring intrinsic chemical reactions during the precursor pyrolysis under inert gas atmosphere, it was revealed that the Ni-N bond formed by a nucleophilic attack of the N atom on the Ni(II) cation center, followed by Ni nucleation below 300 °C, which was promoted by the decomposition of Ni nitride species. The latter was facilitated under the hydrogen-containing atmosphere generated by the NiCl_2_-catalyzed dehydrocoupling reaction. The increase of the temperature to 400 °C led to the formation of a covalently-bonded amorphous Si_3_N_4_ matrix surrounding Ni nanocrystallites.

## 1. Introduction

Transition metal (TM)-based compounds have been widely developed as heterogeneous catalysts in reactions involving energy-relevant transformations [1,2,3]. Growing interest in “sustainable growth”, in industry and science as well as politics, still demands innovation to store, transmit, and/or convert any form of energy; thus, materials discovery appears to be a key element in the innovation cycle of energy conversion, transmission, and storage technologies [4].

Chemical routes toward ceramics are well-suited approaches for designing and synthesizing such highly active catalysts with the required ceramic robustness, especially in harsh environments. A very convenient precursor route is the polymer derived ceramics (PDCs) route, which allows fine control over the chemical composition of the final ceramic materials as well as their phase distribution and nanostructure [5,6,7]. In recent years, TM/Si-based (oxy-)carbide (SiC, SiOC) and carbonitride (SiCN) matrix nanocomposites, such as Ni/SiC, Ni/SiOC, Ni/SiCN(O), and M/SiCN (M = Pd, Ru, Pd_2_Ru, Cu, Ir, Ni, Pt, Co, and Fe) derived from metal-modified polycarbosilanes, polysiloxanes, and polysilazanes have been developed with enhanced catalytic performance and/or higher reusability [8,9,10,11,12,13,14,15,16,17,18,19,20]. Among them, as reported by Kempe et al., polymer-derived Ni/SiCN nanocomposites, while metal Ni is widely accepted as an active catalyst for hydrogenation/dehydrogenation reactions [21,22], are attractive as robust and reusable catalysts to be applied in the catalytic reactions for selective hydrogenation of nitroarenes.

Compared with M/SiCN nanocomposites, the design of M/Si-based nitride nanocomposites systems (i.e., M/Si_3_N_4_) is much more challenging, whereas the silicon nitride matrix could contribute to the catalytic activity of the whole system [23,24,25,26,27,28]. This might be due to the systematically thermodynamically-controlled formation of metal nitride through the reaction of the metal cations chemically-bonded and/or physically-loaded to the polysilazane with ammonia (NH_3_), which is used as extrinsic nitrogen source for the formation of Si_3_N_4_. This is well illustrated in the recent reports on Si_3_N_4_-based nanocomposites, such as titanium nitride (TiN)/Si_3_N_4_ [29,30] and vanadium nitride (VN)/Si_3_N_4_ [31]. However, we recently succeeded in the formation of cobalt (Co)/Si_3_N_4_ nanocomposites through the PDCs route, using PHPS as amorphous Si_3_N_4_ (labeled a-SiN) precursor coordinated with CoCl_2_ as Co source [32], although it was not possible to isolate samples free of ammonium chloride at low temperature while keeping the Si_3_N_4_ matrix amorphous, which is required for catalytic activity [27]. Moreover, the composites were synthesized in flowing NH_3_, for instance, it was impossible to avoid the formation of ammonium chloride as a bi-product via the metal ammine chloride complex formation, especially at lower temperatures, which affected the in situ formation of nanocomposites.

In this study, the polymer-derived Ni/a-SN system has been selected for our synthetic investigation. This is expected to be essential to developing high-performance catalysts for hydrogenation/dehydrogenation reactions. Herein, we report the formation mechanism of Ni nanocrystallites surrounded by amorphous silicon nitride (a-SiN) free of ammonium chloride using PHPS as a-SiN precursor, further coordinated with NiCl_2_ as Ni source. Moreover, inert nitrogen (N_2_) atmosphere was applied for the present precursor pyrolysis to exclude the possibility of reactions of Ni-modified PHPS with atmospheric gas and to precisely monitor the intrinsic reactions during conversion of Ni-coordinated PHPS into Ni/a-SiN nanocomposite. The in situ formation of metallic Ni nanocrystallites within the polymer-derived a-SiN matrix was monitored by using a complete set of characterization techniques, including infrared spectroscopy, thermogravimetric–mass spectrometric (TG-MS) analysis, elemental analysis, powder X-ray diffraction (XRD), transmission electron microscopy (TEM) observation, and Raman spectroscopy. Thus, this study paves the way for the design of Ni/a-SiN nanocomposites, which can be applied in a wide range of catalysis-assisted reactions requiring harsh conditions.

## 2. Materials and Methods

### 2.1. Synthesis of Single Source Precursor for SiNiN System

Commercially available perhydropolysilazane (PHPS, NN120-20, 20 wt% in dibutyl ether solution, Sanwa Kagaku, Corp., Shizuoka, Japan) and nickel chloride (NiCl_2_, 98% purity, Sigma-Aldrich Japan, Tokyo, Japan) were used without further purification. All the reactions and handling of the chemicals for precursor syntheses were carried out under an inert atmosphere of pure argon (Ar), using standard Schlenk line and grove box techniques. The dibutyl ether solvent of as-received PHPS was substituted by super-anhydrous toluene (99.5% purity, Wako Pure Chemical Industries, Ltd., Osaka, Japan) to be 20 wt% for the PHPS contents. Then, the synthesis of Ni-modified PHPS samples was performed at the atomic ratios of Ni in NiCl_2_ to Si in PHPS (Ni/Si) = 0.05 and 0.2. The synthesized precursors were labeled as **0.05NiPHPS** and **0.2NiPHPS**, respectively. In a typical experiment, a 300 mL two-neck, round-bottom flask equipped with a magnetic stirrer was charged with the PHPS (20 wt% toluene solution, 21 mL, 93.1 mmol), then NiCl_2_ (0.564 g, 4.44 mmol, Ni/Si = 0.05) and anhydrous toluene (20 mL) were added to the solution at room temperature. The resulting orange-colored solution was refluxed for 15 h under flowing Ar with stirring. During refluxing, the color of the solution changed from orange to brown-gray. After the reaction mixture was cooled down to room temperature, the solvent was removed under vacuum at 50 °C to give Ni-modified PHPS (**0.05NiPHPS**).

### 2.2. Conversion to Ni/Amorphous SiN Composite

The **0.05NiPHPS** was subsequently pyrolyzed in a quartz tube furnace (Model FUW220PA, Advantec Toyo Kaisha, Ltd., Chiba Japan) under flowing N_2_ (200 mL/min) up to specific temperatures of 200, 300, and 400 °C with the ramping rate of 5 °C min^−1^. Then, each sample was quenched without holding time to afford Ni nanoparticle-dispersed amorphous silicon nitride composite sample labeled as **Ni/SiNX** (X represents the pyrolyzed temperature). As a reference, Ni-free PHPS was also pyrolyzed under the same manner to give amorphous silicon nitride sample denoted as **SiNX**. To study the Ni-nanoparticle formation within the PHPS-derived amorphous Si-N network, pyrolysis of as-received NiCl_2_ alone was performed under flowing NH_3_ or 10% H_2_/Ar (200 mL/min) with the ramping rate of 5 °C min^−1^. When the furnace temperature reached 300 °C, the flowing gas was switched from the reductive gas to N_2_ before quenching as mentioned above.

### 2.3. Characterization

The synthesized single source precursors were characterized by an attenuated total reflection flourier transform infra-red (ATR-FTIR) spectroscopy using FTIR spectrometer (FT/IR-4200IF, JASCO Corporation, Tokyo, Japan) with a diamond prism under an incidence angle of 45° loaded in an ATR attachment (ATR PRO 550S-S/570S-H, JASCO Corporation, Tokyo, Japan). The ATR-FTIR spectra were collected at a resolution of 4 cm^−1^ with a cumulative number of 128 and the spectra were normalized by the strongest peak in the measured region.

To investigate the chemical bonding states, Raman spectra were recorded on pyrolyzed samples and as-synthesized samples (Model inVia, Renishaw, UK) using a single mode solid laser with wavelength 633 nm and power 25 mW for Raman excitation. The Raman spectra were also normalized by the strongest peak.

The thermal behavior up to 800 °C of as-received PHPS and **0.05NiPHPS** was studied by simultaneous thermogravimetric (TG) and mass spectroscopic (MS) analyses (Model STA7200, Hitachi High Technologies Ltd., Tokyo, Japan) coupled with a quadrupole mass–spectrometry (Model JMS-Q1050GC, JEOL Ltd., Tokyo, Japan)]. The measurements were performed under flowing He (100 mL/min) with a heating rate of 10 °C min^−1^.

The chemical composition of the pyrolyzed samples was calculated according to following equation:(1)$$wt\%\left(\mathrm{Si}+\mathrm{Ni}+Cl\right)=100\%-wt\%\left(C\right)-wt\%\left(N\right)-wt\%\left(O\right)$$
where the carbon (C) content was measured using a carbon analyzer (non-dispersive infrared method (Model CS844, LECO Corporation, MI, USA), and the oxygen (O) and nitrogen (N) contents were measured using an oxygen nitrogen hydrogen analyzer (inert gas fusion method, Model EMGA-930, HORIBA, Ltd., Kyoto, Japan), while the Si, Ni, and Cl contents were analyzed by the energy dispersive X-ray spectroscopy (EDS) mounted on a scanning electron microscope (SEM, Model JSM-6010LA, JEOL Ltd., Tokyo, Japan).

The powder X-ray diffraction (XRD) pattern analysis of the pyrolyzed samples was performed on a flat sample stage using Ni-filtered CuKα radiation (Model X’pert, Philips, Amsterdam, The Netherlands). XRD patterns were collected twice with two different scan rates: 4.7 and 0.8° min^−1^ in the regions of 10 to 90° and 35 to 55°, respectively. The average crystallite size (*L*) of metal nickel formed in situ, was obtained from the Scherrer equation as follows:(2)L=Kλ/βcosθ
where λ is the X-ray wavelength in nanometer, K is a constant related to crystallite shape, generally applied as 0.9, and β is the peak width of the diffraction peak at half maximum height in radian. The β in this study was collected from the diffraction peak at 44.5° measured at the scan rate of 0.8° min^−1^.

Transmission electron microscope (TEM) observations and high-angle annular dark-field scanning transmission electron microscope (HAADF-STEM) observations were performed on the pyrolyzed samples in a JEOL JEM-ARM200F operated at an accelerating voltage of 200 kV. The size of the electron probe was approximately 0.1 nm. The convergent angle and the detector collection angle were 22 mrad and 68–280 mrad, respectively. The Ni average size and size distribution histogram were obtained by measuring 50 particles in their longer direction using ImageJ program (NIH free software ver.1.52a).

## 3. Results and Discussion

### 3.1. Chemical Structure of Ni-Modified PHPS

To investigate the chemical bonding states of Ni-modified PHPS, ATR-FTIR spectroscopic analysis was performed on two samples, namely **0.05NiPHPS** and **0.2NiPHPS**, which display the Ni:Si atomic ratio of 0.05 and 0.2, respectively. As-received PHPS—who served as a reference—(black line in Figure 1a) presents typical absorption bands at 3372 (νN-H), 2143 (νSi-H), 1172 (δN-H), and 831 cm^−1^ (νSi-N-Si) [33,34]. Although Ni-modified PHPS samples show similar spectra (red and blue lines in Figure 1a), the relative absorption intensity of δN-H at 1172 cm^−1^ to νSi-N-Si (A_N-H_/A_Si-N-Si_) and that of νSi-H at 2143 cm^−1^ to νSi-N-Si (A_Si-H_/A_Si-N-Si_) decreased with the increase of the Ni/Si atomic ratio (Appendix A). In addition, Ni-modified PHPS samples show shifts of νSi-N-Si band towards the lower wavenumber from 831 to 820–809 cm^−1^, and the shifts increase consistently with the increase of the Ni/Si atomic ratio; thus, with the increase of the Ni content (Figure 1b).

It has been reported that the catalytic formation of bonds between Group 14 and Group 15 elements, such as Si-N bonds resulting from the dehydrocoupling reaction between Si-H and N-H, could be conducted in the presence of various transition metal compounds at the temperature ranges of 20–90 °C [35]. Considering our synthesis conditions (i.e., precursor synthesis conducted in toluene at reflux (~110 °C)), the NiCl_2_-catalyzed dehydrocoupling reactions of Si-H/N-H are expected to proceed in our system to form further Si-N bonds. 

Generally, the frequency of vibration (harmonic-oscillator) is inversely proportional to the square root of the mass of vibrating moiety. For instance, Lucovsky et al. reported that the Si-N stretching band attributed to (≡Si)_3_Si-N= was identified at a lower wavenumber in comparison with (≡Si)_2_HSi-N=, in which one of the silicon moieties (≡Si) was replaced with a H atom [36]. The shift of νSi-N-Si band in Ni-coordinated PHPS toward lower wavenumber is—in this analogy—due to the occurrence of dehydrocoupling reactions catalyzed by NiCl_2_ leading to the formation of ternary silylamino group ((≡Si)_3_N:).

Based on these observations, the Ni-modified PHPS synthesized in this study can be characterized as follows: the ternary (≡Si)_3_N: group formed in situ via the NiCl_2_-catalyzed dehydrocoupling reaction between the (≡Si)_2_HN: group and the H-Si≡ bond is suggested as a possible ligand that coordinates on the Ni(II) cation of NiCl_2_ to afford a complex structure such as a 4-coordinate Ni(II) complex (Figure 1).

### 3.2. Chemical Reaction during in Situ Nano Structuring Process

The occurrence of these mechanisms affects the ceramic conversion of PHPS. Therefore, the thermal behavior of the **0.05NiPHPS** sample has been monitored using TG-MS analysis under flowing inert gas (He). The TG-curve and total ion current chromatogram (TICC) (Figure 2a) indicate that the weight loss contentiously occurs at 150 to 800 °C associated with the evolution of gas species in three temperature ranges as revealed by the simultaneous MS analyses (Figure 2b–e). They show that the weight loss up to 200 °C is due to the evaporation of residual reaction solvent as the evolved species are attributed to toluene (C_6_H_5_CH_3_, m/z = 91, 92, Figure 2b), while at 200 to 800 °C, the weight loss can be divided into two regions based on the dominant species detected:200–480 °C: ammonia (NH_3_, m/z = 16, 17, Figure 2c) and monochlorosilane (SiHCl, m/z = 64, 66, Figure 2d).480–800 °C: hydrochloric acid (HCl, m/z = 36, 38, Figure 2e).

The release of NH_3_ is due to the transamination reaction which is one of the typical cross-linking reactions occurring in polysilazanes [37,38]. Interestingly, the transamination reaction in **0.05NiPHPS** sample, i.e., within the cross-linked PHPS polymer network by the NiCl_2_ modification (Figure 2), was found to proceed approximately 50 °C lower than that in the as-received PHPS sample (Appendix A).

The identification of monochlorosilanes and HCl is most probably due to the reaction between the Si center and/or NH units of PHPS with NiCl_2_ that could form a nitride phase.

To investigate the transformation of Ni-modified PHPS samples into ceramics in more depth, we selected the **0.05NiPHPS** sample to be pyrolyzed at intermediate temperatures (200, 300, and 400 °C) under an inert gas (N_2_) atmosphere. Then, the pyrolyzed intermediates labeled **Ni/SiNX** —with **X** the temperature at which the sample has been exposed—have been characterized by elemental analyses, X-ray diffraction, and Raman spectroscopy.

Table 1 lists the chemical compositions of the pyrolyzed samples (**Ni/SiNX**) along with those of PHPS-derived amorphous silicon nitride samples (**SiNX**) isolated at the same temperatures. At first, the Ni:Si ratio fixed in the early stage of the process, at the polymer level, increased during the pyrolysis indicating the release of Si-containing species, such as monochlorosilane, as identified by TG-MS (Figure 2). To support this observation, the N/Si and Cl/Si atomic ratios in **Ni/SiNX** samples decrease with the pyrolysis temperature increase, which is consistent with the evolution of NH_3_ and monochlorosilane, as shown during TG-MS analysis (Figure 2).

### 3.3. Chemical Composition of Nanocomposites

The XRD analysis reveals that Ni(II) keeps the initial state as dichloride at 200 °C (**Ni/SiN200** sample) since the diffraction pattern composed of the three peaks (2θ = 15.1, 32.8, 52.6°) completely matches the three main peaks detected for the NiCl_2_ used in this study (Appendix A). Then, at 200 to 300 °C, the nucleation of metal Ni (JCPDS No. 01-070-1849, 2θ = 44.5, 51.8, 76.5°) proceeds and the corresponding XRD peaks are associated with those of the secondary phases, such as NH_4_NiCl_3_ (JCPDS No. 00-020-0098, 2θ = 14.7, 33.4, 40.0, 42.9°) and a trace amount of NH_4_Cl (JCPDS No. 01-077-2352, 2θ = 32.8°, **Ni/SiN300** sample). At 400 °C (**Ni/SiN400** sample), the XRD peaks of byproducts disappear and metal Ni becomes the unique single crystalline phase identified in the XRD pattern (Figure 3a,b). It should be noted that Ni nanocrystallite cannot be formed by pyrolysis of pure NiCl_2_ powder under reductive H_2_ or NH_3_ conditions, as well as in inert N_2_ conditions (Appendix A and S5). Therefore, it is strongly suggested that the Ni nanocrystallites are formed in situ through chemical reactions of PHPS with NiCl_2_ leading to the formation of a covalently-bonded amorphous Si_3_N_4_ matrix surrounding Ni nanocrystallites.

The average crystallite size was calculated for the Ni (111) plane by using the Scherrer equation, and the initial average crystallite size at 300 °C of 18.4 nm was evaluated for the **Ni/SiN300** sample, then the crystallite size reached 18.5 nm at 400 °C (**Ni/SiN400** sample).

### 3.4. Mechanistic Investigation of the in Situ Formation of Nanocomposites

Interestingly, FTIR spectroscopy and TG-MS analysis suggested that NiCl_2_ may produce a Ni nitride as a consequence of potential interactions with ((≡Si)_3_N:) or N-H bonds; a Ni nitride compound was not detected as an intermediate by the XRD analysis (Figure 3). Within this context, we conducted further investigation focused on the **0.05NiPHPS** sample, the intermediate states (**Ni/SiN200** and **Ni/SiN300** samples) and the Ni/a-SiN nanocomposite (**Ni/SiN400** sample) and obtained a clear understanding of the in situ formation pathway by the Raman spectroscopic analysis (Figure 4).

The spectra have been recorded for the samples at polymer state as well as those listed in Table 1. At polymer state (**0.05NiPHPS** sample), the spectrum presents two sharp peaks at 169.4 cm^−1^ and 265.1 cm^−1^, which are slightly red-shifted against those of as-received NiCl_2_ at 171.0 cm^−1^ and 266.6 cm^−1^ shown in Appendix A and one additional distinct peak centered at 503.3 cm^−1^ assigned to a Si-Si bond [39,40]. After pyrolysis at 200 °C (**Ni/SiN200** sample), the two peaks due to NiCl_2_ further red-shifted to 168.8 cm^−1^ and 263.9 cm^−1^, respectively.

Feshin et al. reported that the coordination of N(CH_3_)_3_ to GeCl_4_ causes the electron density redistribution to elongate the Ge-Cl bond length and increase the polarity of the Ge-Cl bond [41]. It is generally accepted that the red-shifts of Raman peaks can be interpreted as a decrease in the force constant of chemical bonds, since the frequency is proportional to the square root of the force constant [42]. Therefore, it could be analogically assumed that the N:→Ni dative bond formation makes the Ni-Cl bond of NiCl_2_ weaker and more polar and, hence, more reactive.

Meanwhile, the peak at 503.3 cm^−1^ became broader in the **Ni/SiN200** sample. The deconvolution of the broad signal yields the Si-Si bond peak centered at 498.7 cm^−1^ (**peak 1**) and an additional one at 514.8 cm^−1^ (**peak 2**) assigned to Ni-N bond in Ni_3_N [43]. At 300 °C (**Ni/SiN300** sample), the two peaks due to NiCl_2_ disappear and the peak area ratio of **peak 2**/**peak 1** decreases from 1.58 (**Ni/SiN200** sample) to 1.02 (**Ni/SiN300** sample), then at 400 °C, the **peak 2**, attributed to the Ni-N bond, disappears and the **Ni/SiN400** sample presents the **peak 1,** due to the Si-Si bond, as a single peak (Figure 4a). It should be noted that Raman spectrum of the **Ni/SiN400** sample in the wider Raman shift range is free from typical peaks attributed to amorphous silicon oxides such as Si-O-Si (820 cm^−1^) and Si-(OH)_x_ (920 and 1079 cm^−1^) [44], while other peaks are found to correspond to the Raman peaks detected for SiN400 and thus attributed to PHPS-derived Si-N matrix networks (Appendix A). These results are consistent with the result of elemental analysis, showing that oxygen impurity is very low in both samples.

Under the present pyrolysis condition up to 400 °C, it is important to note that NiCl_2_ keeps its initial state without thermal decomposition (Appendix A). Thus, the results obtained by the TG-MS, XRD, and Raman spectroscopic analyses reveal that the reaction of the activated NiCl_2_ with PHPS begins to start below 200 °C and the subsequent metal Ni nucleation and crystallization occurs between 200 to 300 °C. Moreover, it is strongly suggested that the in situ formation of metal Ni proceeds via the formation of Ni nitride species as intermediates. The Raman spectroscopic analysis reveals that Ni_3_N is one possible intermediate phase (Figure 4a), which is well consistent with the reported thermodynamic data: Ni_3_N has the lowest formation energy of NiN_x_ (x = 0 to 1/3) [45] and is a thermodynamically favorable phase at lower temperatures in the phase diagram for Ni-N system [46].

Regarding the Ni-N bond formation, since the temperature range detected by the Raman spectroscopic analysis was as low as 200 to 300 °C, the direct nitridation of Ni by atmospheric N_2_ is excluded. Therefore, the Ni-N bond is intrinsically formed by the chemical reaction of NiCl_2_ with PHPS as the nitrogen source via ternary silylamino groups.

It should be noted that the formation of Si-Si bond is reasonable since the N/Si atomic ratio of all the **Ni/SiNX** samples is lower than 1 (Table 1); indeed, all the **SiNX** samples with the N/Si atomic ratio <1 (Table 1) exhibited the typical Raman peak attributed to Si-Si bond (Figure 4b).

Based on these results, reactions during the polymer to metal Ni/a-SiN conversion process can be suggested as follows (Figure 2): After the chemical modification of PHPS with NiCl_2_ to afford Ni complexes (**1**), Ni-N bond formation would proceed via the nucleophilic attack of N atom of the silylamino group on the center Ni(II) cation to replace Cl. Since the N:→Ni dative bond has been already formed, the N atom would be easily accessible to Ni(II), and thus a S_N_2 reaction via the transition state (**2**) is proposed. Subsequently, the nucleophilic attack of the released chloride anion (Cl^–^) on other electrophiles, such as Si center of PHPS moieties (**3**), would proceed, leading to the evolution of monochlorosilane, as detected by the TG-MS analysis. Then, the formation of Ni nitride species (**5**) could proceed through the thermal decomposition of in situ preformed species, having Ni-N bond (**4**).

In general, under N_2_ or NH_3_ atmosphere, Ni_3_N, which is considered as the possible intermediate in this work, starts decomposing thermally at about 400 °C, accompanied by the formation of metallic Ni and gaseous nitrogen, while the Ni_3_N decomposition temperature has been found to be as low as 157 °C due to its chemical fragility under an H_2_ atmosphere [47,48]. Thus, it could be speculated that the local hydrogen partial pressure around the Ni nitride species embedded within the polymer-derived amorphous Si-N network reached a sufficiently high level to promote the immediate decomposition of the Ni nitride species (**5**), followed by the nucleation and subsequent crystallization of metal Ni at 200 to 300 °C, shown as (**6**) and (**7**) in Figure 2. As previously mentioned, NiCl_2_ exhibited high catalytic activity to facilitate Si-H/N-H dehydrocoupling reactions in PHPS moieties. It should be noted that this NiCl_2_-catalyzed dehydrocoupling reaction proceeded contentiously during pyrolysis, which facilitated the polymer to the ceramic conversion of PHPS up to 400 °C.

In addition, the transamination reaction leads to the formation of the byproducts, NH_4_NiCl_3_ and NH_4_Cl, as shown in Figure 3, resulting from the nucleation reactions of the gaseous NH_3_ formed in situ with Ni complexes and the NH_3_ with HCl formed in situ, respectively. Then, above 300 °C, the byproducts thermally decompose to yield gaseous NH_3_ and HCl, while the regenerated NiCl_2_ species would participate in the Ni-N bond formation reaction as shown in Figure 2.

To assess the phase nanostructure of the metal Ni/amorphous silicon nitride composites in more detail, TEM observation was performed on the **Ni/SiN400** sample. As shown in Figure 5a, many dark dots correspond to metallic Ni crystallites embedded within amorphous silicon nitride matrix. In fact, the fringe spacing calculated from Figure 5b is found to be 0.125 nm, 0.177 nm, and 0.202 nm, corresponding to the (220), (200), and (111) lattice planes of fcc Ni (JCPDS No. 01-070-1849), respectively.

A brief size distribution analysis by measuring the size of 50 Ni nanocrystallites revealed a relatively wide range of size distribution from 7.32 to 43.4 nm; however, most of the Ni crystallites were below 25 nm in size, and the mean size was found as 18.0 nm. This is confirmed by the high-resolution image (Figure 5c) which reveals the presence of a nanocrystal with a lattice spacing of 0.202 nm, corresponding to the *d*-spacing of the lattice plane of the Ni structure, i.e., the (111) direction of the *fcc* cubic Ni structure which agrees with the phase identified, as well in the X-ray diffractogram (Figure 3), as seen in the SAED pattern (Figure 5b) of the same sample.

## 4. Conclusions

In this study, we reported on the in situ formation of metallic nickel nanocrystallites, with an average Ni size of 18.0 nm, finely dispersed within a polymer-derived amorphous silicon nitride matrix at pyrolysis temperatures as low as 400 °C. The results can be summarized as follows:ATR-FTIR spectroscopic analysis revealed that NiCl_2_ had high catalytic activity for dehydrocoupling reactions between Si-H and N-H of PHPS under the present precursor synthesis condition conducted in toluene at reflux (~110 °C), leading to the formation of ternary silylamino groups ((≡Si)_3_N:). Consequently, the ternary silylamino group coordinated the Ni(II) cation of NiCl_2_ to afford a complex such as the 4-coordinated Ni(II) complex.TG-MS and Raman spectroscopic analyses revealed that the Ni-N bond in Ni nitride species is intrinsically formed via the S_N_2 reaction of the preformed 4-coordinated Ni(II) complex at 200 °C: the nucleophilic attack of the N atom of the silylamino group on the center Ni(II) cation and simultaneous elimination of Cl^–^ as a leaving group. Subsequently, the nucleophilic attack of the released Cl^–^ on other electrophiles, such as Si center of PHPS moieties, would proceed accompanied by the evolution of monochlorosilane, while the N-bonded Ni species subsequently decomposed to give the Ni nitride species.XRD, ATR-FTIR, and Raman spectroscopic analyses revealed that Ni nanocrystallites started to form at temperatures as low as 200 to 300 °C through the decomposition reaction of the in situ formed Ni nitride species facilitated by H_2_, which was generated through the NiCl_2_-catalyzed dehydrocoupling reaction of PHPS. In addition, this NiCl_2_-catalyzed dehydrocoupling reaction of PHPS accelerated the polymer to the ceramic conversion of PHPS up to 400 °C.

Therefore, it can be concluded that the low temperature in situ formation of Ni/amorphous silicon nitride composites was governed by the synergistic effect of the following two reactions attributed to the high catalytic performance of NiCl_2_ for dehydrocoupling reaction of PHPS: (i) the Ni(II) complex formation allows in situ formation of the Ni nitride species as an intermediate below 200 °C and (ii) the immediate decomposition of the Ni nitride species promotes Ni nucleation at 200 to 300 °C.

Based on our results obtained in this study, it is highly expected that the ability to control material properties such as the size of Ni nanocrystallites, chemical composition, and pore size distribution by designing polymeric precursors can be achieved.

Thus, further investigation on the molecular structure of single source precursors for the ideal metal/nitride nanocomposites is in progress and the nanocomposites are expected to be applied as a new, highly active catalyst for important reactions, such as the hydrogenation of CO_2_, which is a promising way to solve the environmental and energy affairs caused by CO_2_ emissions. Such a study is under investigation and will be published separately.

## Data Availability

The data reported in this study are available from the authors upon reasonable request.

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
