# Peer review of "Mechanistic Investigation of the Formation of Nickel Nanocrystallites Embedded in Amorphous Silicon Nitride Nanocomposites"

_nanomaterials, 2022, doi:10.3390/nano12101644_

Round 1

Reviewer 1 Report

The authors investigated the formation of nickel nanocrystallites embedded in amorphous silicon nitride nanocomposites. It was revealed that Ni-N bond formed by a nucleophilic attack of N atom to Ni(II) cation center, followed by Ni nucleation below 300 °C, which was promoted by the decomposition of Ni nitride species. The publication of this manuscript will provide guidance for the researchers to develop Ni nanocrystallites with chemical composition and pore size distribution well controlled, which are expected to be applied as new highly active catalyst for important reactions such as hydrogenation of CO2 and improve the environment. To present a high-quality publication, following revisions are advised:

  1. Density functional theory method has been applied to investigated the growing mechanism of metal nanoparticles. This method should be mentioned in the introduction part. Please refer to Energy Storage Mater. 34 (2021) 107-127, Energy Storage Mater. 27 (2020) 279-296.
  2. The inset in Fig. 5a is not sharp. Please improve it.
  3. The language of English should be improved. There are some spelling and grammar mistakes through the manuscript.

Author Response

Dear Reviewer 1#

Please find an attached file.

Thank you for the valuable comments. We have considered all of your comments and revised our manuscript with respect to your suggestion. Your valuable suggestions and comments were carefully considered during the manuscript revision. All changes performed in the text are highlighted using yellow marker.

Reviewer 2 Report

This work conducted the mechanistic study on the formation of nickel (Ni) nanocrystallites during the formation of amorphous silicon nitride (Si3N4) using NiCl2 coordinated perhydropolysilazane (PHPS) under the hydrogen containing atmosphere at 400 C. The work is interesting, lies within the scope of the journal and could find wide readerships. The manuscript is well-presented and organized. However, before acting, I suggest the revisions on the following points.

  • What was the percentage composition of Ni and Si3N4?
  • How were the surface oxidation states of the elements particularly of Si3N4?
  • In the conclusion part, the authors stated that “By controlling molecular structures of Ni(II) complexes, we can control material properties such as the size of Ni nanocrystallites, chemical composition and pore size distribution.” However, no supporting data on this could be find in the manuscript.

Author Response

Dear Reviewer 2#

Please find an attached file.

Thank you for the valuable comments. We have considered all of your comments and revised our manuscript with respect to your suggestion. Your valuable suggestions and comments were carefully considered during the manuscript revision. All changes performed in the text are highlighted using yellow marker.

Reviewer 3 Report

In this research the authors investigated the formation of Ni nanocrystallites within polymer-derived amorphous silicon nitride matrix in a pyrolysis process by using some characterizations such as ATR-FTIR, Raman spectra, TG-MS, SEM, EDS, and XRD. The research is interesting and I will recommend its publication after major revision. Some comments can be shown as follows:

  • In the introduction, the authors should importantly discuss what difference for the preparation between Co/Si3N4 and Ni/Si3N4 so that the readers can understand how and why to get the resulting Ni/Si3N4 in the present work. It is not necessary to spend one big paragraph on the discussion of the chemical routes toward ceramics.
  • XPS should be further conducted to check Ni and Si3N4 by valence.
  • Figure 5: the insert of Figure 5(a) is not clear. All figures, especially Figures 2 and 3, should be adjusted according to the requirement of the honored Nanomaterials.

Author Response

Dear Reviewer 3#

Please find an attached file.

Thank you for the valuable comments. We have considered all of your comments and revised our manuscript with respect to your suggestion. Your valuable suggestions and comments were carefully considered during the manuscript revision. All changes performed in the text are highlighted using yellow marker.

Round 2

Reviewer 2 Report

The authors have carefully addressed all my concerns. In my views, the revised manuscript can now be accepted.

Reviewer 3 Report

The authors have addressed my questions. I recommend this present manuscript can be accepted and published in the honored Nanomaterials.